# The Important Role of m6A-Modified circRNAs in the Differentiation of Intramuscular Adipocytes in Goats Based on MeRIP Sequencing Analysis

**DOI:** 10.3390/ijms24054817

**Published:** 2023-03-02

**Authors:** Jianmei Wang, Xin Li, Wuqie Qubi, Yanyan Li, Yong Wang, Youli Wang, Yaqiu Lin

**Affiliations:** 1Key Laboratory of Qinghai-Tibetan Plateau Animal Genetic Resource Reservation and Utilization, Ministry of Education, Southwest Minzu University, Chengdu 610041, China; 2Key Laboratory of Qinghai-Tibetan Plateau Animal Genetic Resource Reservation and Exploitation of Sichuan Province, Southwest Minzu University, Chengdu 610041, China; 3College of Animal & Veterinary Science, Southwest Minzu University, Chengdu 610041, China

**Keywords:** MeRIP-seq (m6A-seq), intramuscular adipocyte, goat, m6A-circRNAs

## Abstract

Intramuscular fat contributes to the improvement of goat meat quality. N^6^-Methyladenosine (m6A)-modified circular RNAs play important roles in adipocyte differentiation and metabolism. However, the mechanisms by which m6A modifies circRNA before and after differentiation of goat intramuscular adipocytes remain poorly understood. Here, we performed methylated RNA immunoprecipitation sequencing (MeRIP-seq) and circRNA sequencing (circRNA-seq) to determine the distinctions in m6A-methylated circRNAs during goat adipocyte differentiation. The profile of m6A-circRNA showed a total of 427 m6A peaks within 403 circRNAs in the intramuscular preadipocytes group, and 428 peaks within 401 circRNAs in the mature adipocytes group. Compared with the intramuscular preadipocytes group, 75 peaks within 75 circRNAs were significantly different in the mature adipocytes group. Furthermore, the Gene Ontology (GO) and Kyoto Encyclopedia of Genes and Genomes (KEGG) analyses of intramuscular preadipocytes and mature adipocytes showed that the differentially m6A-modified circRNAs were enriched in the PKG signaling pathway, endocrine and other factor-regulated calcium reabsorption, lysine degradation, etc. m6A-circRNA–miRNA–mRNA interaction networks predicted the potential m6A-circRNA regulation mechanism in different goat adipocytes. Our results indicate that there is a complicated regulatory relationship between the 12 upregulated and 7 downregulated m6A-circRNAs through 14 and 11 miRNA mediated pathways, respectively. In addition, co-analysis revealed a positive association between m6A abundance and levels of circRNA expression, such as expression levels of circRNA_0873 and circRNA_1161, which showed that m6A may play a vital role in modulating circRNA expression during goat adipocyte differentiation. These results would provide novel information for elucidating the biological functions and regulatory characteristics of m6A-circRNAs in intramuscular adipocyte differentiation and could be helpful for further molecular breeding to improve meat quality in goats.

## 1. Introduction

Goat is one of the most widely consumed meats in the world. Compared with beef or pork, goat meat is an important source of high-quality protein, healthy fats, low-calorie intramuscular fats and saturated fats [1], and plays a crucial role in human nutrition. Meanwhile, studies have reported that increased intramuscular fat (IMF) content can improve meat quality significantly in pigs [2]. Therefore, IMF, as a main form of fat deposition, is an important factor affecting meat quality traits, such as its tenderness, juiciness and taste, making it an economically important factor in goat breeding. For meat, the content of IMF is an important trait influencing meat quality, and the differentiation of preadipocytes is a key factor affecting IMF deposition [3,4]. Thus, there are intensive efforts for exploring the molecular mechanisms underlying IMF deposition, which is of great significance for improving the quality of goat meat. In recent years, increasing evidence has suggested a potential role for non-coding RNAs (ncRNAs) in IMF deposition at the post-transcriptional level [5].

Circular RNAs (circRNAs), a novel class of endogenous ncRNAs that form covalently closed-loop structures and lack a 5′ cap or 3′ poly-A tail [6], have been widely detected in eukaryotes [7]. Previous studies have reported that circRNAs can participate in various physiopathologic processes by mediating protein–RNA interactions [8,9], such as miRNA or protein sponges [10], or modulating protein translation [11], mostly by acting as competing endogenous RNA (ceRNA) to relieve suppression. With the continuous exploration of its diverse functions, circRNAs have been shown to regulate various biological processes extensively, including fat deposition. However, studies on how circRNAs are regulated before exerting specific biological functions are still limited.

m6A is a common epitranscriptomic modification of RNA, which has been found to affect the metabolism of messenger RNAs (mRNAs), including splicing, export, translation, and decay, and plays vital roles in the functions of various ncRNAs, such as long noncoding RNAs (lncRNAs), microRNAs, circRNAs, small nuclear RNAs (snRNAs), and ribosomal RNAs (rRNAs) [12,13]. Interestingly, circRNA can be regulated by m6A modification, showing a different m6A pattern from that of mRNA [14]. Recent studies have shown that m6A-modified circRNAs have been associated with diseases [15,16]. Meanwhile, Hui et al. [17] believed that m6A-modified circRNAs were involved in secondary hair follicle (SHF) development and cashmere growth in goats. Unfortunately, there are currently no reports on circRNA m6A modification in the intramuscular adipocyte differentiation of meat goats.

Therefore, to further identify the potential function of m6A modification in regulating circRNA, our objective was to explore the regulatory differences in m6A methylation that mediate circRNA translation in the intramuscular adipocytes of meat goats before and after differentiation using MeRIP-seq (m6A-seq) sequencing technology. The findings provide new knowledge to understand the regulatory mechanisms of adipocyte differentiation and fat deposition in meat goats.

## 2. Results

### 2.1. Identification of an Intramuscular Preadipocyte Differentiation Model in Goats

In order to examine the fat deposition and lipid droplet morphology in the cultured intramuscular adipocytes, we performed Oil Red O and BODIPY staining. After 3 days of induction, lipid droplets could be observed with Oil Red O and BODIPY staining (nuclei were counterstained with DAPI) (Figure 1), which could indicate that intramuscular preadipocytes (IMPA) and adipocytes (IMA) models were successfully established.

### 2.2. Overview of the circRNA-seq and MeRIP-seq Data

To investigate the circRNA profile in goat intramuscular adipocytes before and after differentiation, purified cellular RNA was subjected to circRNA-seq (m6A-seq input library) and MeRIP-seq. The sequencing raw reads were generated from the IMPA group and the IMA group. With three biological replicates, the circRNA-seq and MeRIP-seq sequencing of 12 libraries generated a total of 289.49 Gb of data, with each library averaging from 12.98 Gb to 14.22 Gb of data. The Q30 results in each library were >93.82%, and the GC percentage was less than 59%, as listed in Table 1. Subsequently, more than 95.30% of the clean reads were perfectly mapped to the goat reference genome (assembly ARS1, https://www.ncbi.nlm.nih.gov/genome/?term=goat (accessed on 11 November 2021)), and 83.78~91.49% uniquely mapped reads were obtained from the total mapped reads from the 12 samples (Table 2). The goat genome and circRNA-seq and MeRIP-seq data information are provided in Appendix A. The experimental strategy is shown in Figure 2.

### 2.3. Characteristics of m6A-Modified circRNAs in the Intramuscular Adipocytes of Goats before and after Differentiation

We used circRNA-seq to compare the differences in circRNAs between intramuscular preadipocytes (IMPA) and adipocytes (IMA) in goats. We found that most circRNAs were between 200–700 bp and derived from sense-overlapping RNAs (Appendix A). Moreover, circRNAs were mainly distributed on chromosome 7 (Appendix A).

The MeRIP-seq data for m6A in the IMPA and IMA groups were compared and analyzed; there were 427 m6A methylation peaks within 403 circRNAs in the IMPA group, and 428 peaks within 401 circRNAs in the IMA group. According to the differences and overlaps in m6A-modified circRNA transcripts, 64 methylation peaks and 63 circRNAs were uniquely modified in the IMPA group, and 65 methylation peaks and 61 circRNAs were uniquely modified in the IMA group. In addition, 363 peaks were consistently observed in the two groups, and 340 circRNAs within both groups were modified by m6A (Figure 3A,B).

Further analysis was performed to assess the features of m6A-modified circRNAs. The number of m6A methylation peaks in each circRNA was highly similar in the IMPA and IMA groups (Figure 3C). We found that almost 61.94% of methylated circRNAs hold only one m6A peak, and most circRNAs contain one to three m6A peaks, which indicates that m6A modification sites are not unique in circRNAs. Moreover, the length results of m6A-modified circRNAs in each group showed that the length of most m6A-modified circRNAs were less than 2000 bp, and the length characteristics of the two groups were similar (Figure 3D). The sources of m6A-circRNAs were most correlated with sense-overlapping RNAs (Figure 3E). Finally, chromosome distribution also revealed that m6A-methylated circRNA is more likely to be present on chromosome 7 (Figure 3F).

### 2.4. Differential Expression of m6A-Modified circRNAs in Different Goat Adipocytes 

Based on a *p* value < 0.05 and |Log_2_ (fold change)| > 1.5, 75 m6A methylation peaks within 75 circRNAs were screened out between the IMPA and IMA group. Among them, 44 hypermethylated peaks were within 44 circRNAs (e.g., circRNA_NUCB1), and 31 hypomethylated peaks were within 31 circRNAs (e.g., circRNA_ZMYND8), as seen from Appendix A. Data visualization analysis was performed using IGV to show the differential m6A peaks between the IMPA and IMA groups (Figure 4A). The top 10 differentially methylated circRNAs with hypermethylation or hypomethylation in the IMA group compared to the IMPA group are shown in Table 3. Meanwhile, the expression profiling was identified by hierarchical clustering analysis, confirming that undifferentiated and differentiated cells exhibited dramatically differentially expressed methylation circRNAs profiles (Figure 4B). GO and KEGG pathway enrichment analyses for the differentially m6A-modified circRNA source genes were performed (*p*-value < 0.05). GO annotation of m6A-modified circRNAs illustrated that they were mainly enriched in cytoplasm, nucleus and metal ion binding (Figure 4C). KEGG analysis indicated that they were enriched in the PKG signaling pathway, endocrine and other factor-regulated calcium reabsorption and lysine degradation (Figure 4D).

### 2.5. Regulatory Network of the Differential m6A-circRNAs between IMA and IMPA Groups

In recent years, studies have found that circRNAs are able to regulate the expression of target genes as sponges for miRNAs based on complementary base pairing. By predicting the target miRNA for both the circRNA and mRNA, we constructed a m6A-circRNA–miRNA–mRNA ceRNA interaction network. In this study, according to a max-score > 150 and max-energy < −30, a total of 12 hypermethylated circRNAs, 14 miRNAs and 55 mRNAs (Figure 5A), and 7 hypomethylated circRNAs, 11 miRNAs, and 22 mRNAs were identified in IMA and IMPA groups (Figure 5B). These associations of m6A-circRNA–miRNA–mRNA interactions are shown in detail in Appendix A. In the interaction network of the two groups, many fats deposition and lipid metabolism miRNAs were predicted, such as miR-103-5p, miR-423-5p and miR-423-3p. In addition, we also found that m6A-modified circRNAs exhibited several m6A-circRNA –miRNA–mRNA regulatory pathways. For instance, m6A-circRNA _1659 may sponge two miRNAs (miR-33b-3p and miR-18a-3p) to further individually or cooperatively regulate the expression of their target genes through a ceRNA network mechanism (Figure 5B).

### 2.6. Conjoint Analysis of circRNA-Seq and MeRIP-Seq

To further explore the potential function of circRNAs with m6A modification in the IMPA and IMA groups, a conjoint analysis of circRNA-seq and MeRIP-seq was performed. Based on a *p* value < 0.05 and |Log_2_ (fold change)| > 1.5, 450 differentially expressed circRNAs were detected in the two groups, including 263 upregulated circRNAs and 187 downregulated circRNAs (Figure 6A). Simultaneously, we constructed a clustered heat map to further explore the potential roles of the circRNAs (Figure 6B). Moreover, GO ontology and KEGG pathway analyses were performed to analyze the differentially expressed circRNAs. The GO analysis of the differentially expressed circRNAs illustrated that the meaningful terms (*p* < 0.05) may be related to lung development, protein autophosphorylation, striated muscle myosin thick filament, etc. (Appendix A). The KEGG enrichment showed that the top 10 significantly enriched signaling pathways were enriched based on the significantly differentially expressed circRNAs in these two groups (*p* < 0.05). These pathways included the MAPK signaling pathway, tight junction, lysine degradation, FoxO signaling pathway, cGMP-PKG signaling pathway, etc. (Appendix A). Furthermore, the correlation between the differentially m6A-modified circRNAs and the corresponding circRNA expression levels was analyzed by combining MeRIP-seq and circRNA-seq. There are 20 significantly upregulated circRNAs with hypermethylation (2 annotated genes and 18 unannotated genes), 2 downregulated circRNAs with hypomethylation (1 annotated gene and 1 unannotated gene) and 1 upregulated circRNA with hypomethylation that were found in the preadipocyte and adipocyte groups (Figure 6C).

### 2.7. Verification of circRNA Expression Profiles Using qRT-PCR

To verify the reliability of circRNA-seq results, four candidate circRNAs were randomly selected from the differentially methylated circRNAs obtained from the screening, and UXT was used as an internal reference for qRT-PCR analysis. The results showed that circRNA_PAPD7 and circRNA_LMO7 were significantly upregulated during the differentiation of intramuscular adipose cells. On the other hand, circRNA_SP3 and circRNA_CHD9 were significantly downregulated during the differentiation of intramuscular adipose cells. These results were consistent with the circRNA-seq trend, indicating the credibility of the circRNA-seq results (Figure 7).

## 3. Discussion

Fat deposition is a very important economic trait that determines goat production, feed efficiency and meat quality, including flavor and tenderness. Studies have shown that the differentiation of intramuscular lipid deposition is a complex biological process regulated by multiple genes, signal pathways and transcription factors [18,19,20,21]. Thus, elucidation of the molecular mechanism underlying meat quality traits in goats will have both biological and economic consequences.

Over the past few years, increasing lines of evidence indicate that m6A modification in circRNA molecules plays significant roles in various cells [22,23,24]. Nevertheless, the potential roles of m6A-modified circRNA in most livestock, and especially in the differentiation of goat intramuscular preadipocytes, has remained largely unclear. To the best of our knowledge, our study is the first to screen for m6A-modified circRNA in goat preadipocytes and adipocytes using MeRIP-seq technology. CircRNAs were generated by back splicing of pre-mRNAs through different pathways. It has been confirmed that the major source of circRNAs is derived from exons and exists in a large number of eukaryotic cells [25,26,27]. Some scholars have shown that circRNAs are generated by contranscription and competition with conventional splicing [28]. However, our results indicated that the characteristics of m6A-modified circRNAs changed before and after differentiation of intramuscular adipocytes. Most of the differential m6A-modified circRNAs are longer and come from sense-overlapping regions, which means that these differential and long m6A-modified circRNAs derived from sense-overlapping regions play a more important function, providing new insights into the regulatory mechanism of m6A-modified circRNAs of different adipocytes.

In the present study, approximately 75 differentially m6A-modified circRNAs were identified before and after the differentiation of goat intramuscular adipocytes. GO subcategory analysis revealed that they were mainly enriched in the cytoplasm, nucleus and metal ion binding. The KEGG enrichment analysis based on the differentially m6A-modified circRNAs demonstrated that the PKG signaling pathway, endocrine and other factor-regulated calcium reabsorption and lysine degradation play a vital role in the adipose differentiation.

Maimaitiyiming et al. [29] suggested that increased PKG signaling stimulates brown adipocyte differentiation, promotes healthy expansion and browning of white adipose tissue, and stimulates white adipose tissue lipolysis. Endocrine and other factor-regulated calcium reabsorption is related to the immune system, and it has a critical role in adipose differentiation [30]. In addition, it has been shown that the dietary lysine to energy ratio mainly determines the rate of protein and fat deposition [31]. Yang et al. [32] demonstrated that lysine degradation plays a promoting role in the process of fat differentiation, which is conducive to fat deposition. These are consistent with our results. Therefore, we conclude that the different m6A-modified circRNAs might be involved in the differentiation of intramuscular preadipocytes.

In recent years, a growing number of studies have found that circRNA can be used as a molecular sponge to interact with miRNA to regulate mRNA [33,34]. By integrating the data from the analyses of circRNAs, miRNAs, and mRNAs, hub ceRNAs networks were constructed for goat adipogenic differentiation. In our ceRNA network, we found that 11 downstream genes in the ceRNA pathways were strongly related to candidate m6A-modified circRNAs in the present study, suggesting that these circRNAs might play functional roles during adipogenesis. It is worth noting that fibronectin type III domain-containing protein 3B (*FNDC3B*) regulates white fat browning and adipogenesis [35]. Moreover, *TTN* (Titin) has been related to changes in intramuscular fat deposition, possibly by exerting effects on adipocyte lineage cells or on the milieux surrounding them [36]. In our study, we found that miR-103-5p was able to regulate *TTN*, *ZNF536* and *WDR76* in three ceRNA networks, and multiple circRNAs had binding sites with miR-2305. Thus, we speculated that circRNA_1944 (circFNDC3B) and circRNA_0582 (circTTN) potentially regulate goat adipogenesis. However, in-depth studies on the functions of goat circFNDC3B, circTTN, circRNA_0582 (circZNF536) and circRNA_0582 (circWDR76) on adipogenic differentiation are essential. Previous research reported that *LAMA5*, *HDAC11*, *CCND2*, *EBF3* were associated with adipogenesis and fat deposition [37,38,39]. We found that circRNA_1689 might influence adipogenic differentiation by regulating downstream genes (*LAMA5* and *EBF3*) through two miRNAs (miR-874-3p and miR-874-3p), and that circRNA_0873 might influence adipogenic differentiation by regulating downstream genes (*HDAC11* and *CCND2*) through one miRNA (miR-1343). Based on the above results, we believe that the m6A-circRNAs, as a “molecular sponge” of these miRNAs, may play essential roles in establishing an optimal expression balance of their target genes during goat adipocyte differentiation, in which m6A modifications may be required, as the m6A-circRNA plays an important role in regulating the proliferation and differentiation of adipocytes and myocytes.

To further reveal the relationship between circRNA and m6A modification, we performed an analysis of circRNA-seq. We found a total of 450 circRNAs with expression differences. In the conjoint analysis of MeRIP-seq, we found that a total of 23 circRNAs showed a significant association between expression and m6A modification; of these, 3 were annotated genes and 20 were unannotated genes. Earlier studies have indicated that m6A modification is closely related to circRNA expression [40]. For instance, Zhang et al. suggested that circRNA accumulation is associated with enhanced splicing at the m6A site and m6A modification may interfere with sperm motility by influencing circRNA expression levels [41]. In the present study, we showed that the expression of two m6A-circRNAs, including circRNA_0873 (circRNA_SLC8A3) and circRNA_1161 (circRNA_DEPTOR), were dramatically upregulated in adipocytes as compared to preadipocytes, and the majority of m6A-circRNAs were expressed at a medium level with a positive relationship between circRNA expression and m6A methylation modification. Thus, it can be suggested that these two m6A-circRNAs (circRNA_0873 and circRNA_1161) may be implicated in the physiological process of goat adipocyte differentiation by constituting coordinated regulatory pairs. In this process, the m6A modifications within the circRNAs might play an important role in promoting the differentiation of goat adipocytes. Additionally, the four circRNAs in the comparison of IMF before and after differentiation were verified by qPCR, and the results were basically consistent with those of RNA-seq. This shows that our RNA-seq discovery is reliable. Based on sequencing data, we considered that these circRNAs play a role in the intramuscular adipocytes of goats before and after differentiation. Although these newly identified circRNAs have not been reported in studies of intramuscular adipocyte differentiation, they can provide some preliminary data for further study.

## 4. Materials and Methods

### 4.1. Isolation and Cell Culture of Goat Intramuscular Preadipocytes

Goat intramuscular preadipocytes were collected from the longissimus dorsal muscle of 7-day-old Jianzhou Daer goats (*n* = 3) (Sichuan Jianyang Dageda Animal Husbandry Co., Ltd., Sichuan, China). The intramuscular preadipocytes were isolated and cultured as described by Xu et al. [42].

### 4.2. Preadipocyte Differentiation Induction

DMEM/F12 (Hyclone, Logan, UT, USA) containing 10% FBS (Hyclone, Logan, UT, USA) and 50 μmol·L^−1^ oleic acid (Sigma, St. Louis, MO, USA) induced differentiation of goat intramuscular adipocytes, and cells were collected at 0 and 3 days [4,43].

### 4.3. Oil Red O and BODIPY Staining

The Oil Red O staining and BODIPY staining were used to distinguish mature adipocytes from preadipocytes during the process of culture. The Oil Red signal was quantified by measuring the absorbance at 490 nm (OD 490) as a semi-quantitative assessment method to determine the extent of differentiation. The fluorescence intensity of the BODIPY signal (arbitrary units, in %) was analyzed using the ImageJ tool (NIH, Bethesda, MD, USA).

### 4.4. RNA Extraction, Library Construction, and Sequencing

Total RNA from 6 samples was extracted. We have generally utilized 100 ng of RNA for library construction for MeRIP-circRNA sequencing. Briefly, the mRNA with polyA in the total RNA was enriched by Oligo-dT magnetic beads. The intact mRNA was then fragmented using an ultrasound machine. The segmented RNA was divided into two parts. One part was added to an m6A-capturing antibody to enrich the mRNA fragments containing m6A methylation (MeRIP-seq), and the other part was used as an Input to directly construct a conventional transcriptome sequencing library (circRNA-seq). The m6A antibody was enriched by magnetic beads, and the mRNA fragments containing m6A were recovered. The conventional sequencing library was constructed according to the transcriptome library construction process. Illumina Hiseq X Ten was used for high-throughput sequencing of the library.

### 4.5. Sequencing Data Analysis

After paired-end sequencing, raw data were first filtered according to Q30 and GC content; fastp software (v0.20.0) was used to obtain high-quality reads. Hisat2 software (v2.1.0) was used to align high-quality reads to the goat reference genome, CIRI2 software (v2) was used for circRNA detection and identification, and the MeTDiff software was used for methylation peak calling and differential peak identification. The circBase database and Circ2Traits were used to annotate the identified circRNA. Then, DESeq2 software (v1.14.1) was used for data standardization and differentially expressed circRNA screening (log_2_FC ≥ 1.5, *p*-value ≤ 0.05).

### 4.6. Bioinformatics Analysis and Statistical Analysis

The DAVID database was used to conduct GO enrichment analysis [44]. KOBAS software (http://kobas.cbi.pku.edu.cn (accessed on 14 February 2023)) [45] was used to test the statistical enrichment of differentially expressed circRNA source genes in KEGG pathways. A *p* value < 0.05 was considered significant. The R language (v1.42.0) and related packages were used to visualize the results.

The m6A-circRNAs/miRNA interactions were predicted using miRanda (http://www.microrna.org/ (accessed on 16 September 2022)) [46]. miRanda was used to predict the downstream mRNA targets of the predicted miRNA. The R language (v1.42.0) and related packages were used to visualize the results.

Additionally, statistical analysis was conducted with the SPSS 17.0 program (SPSS Inc., Chicago, IL, USA). Results are shown as the mean ± SEM and the data are representative of three biological and two technical replicates. * *p* < 0.05, ** *p* < 0.01.

### 4.7. Validation of Gene Expression by RT-qPCR Technique

Primers were designed using Primer-BLAST on the NCBI website (Table 4). First-strand cDNA was synthesized using a reverse transcription system (Takara, Shiga, Japan) according to the manufacturer’s instructions, and the cDNA was used for quantitative real-time PCR, which was carried out with the SYBR Prime Script RT-PCR Kit (Takara, Shiga, Japan). *UXT* was used as the housekeeping gene for normalization of the gene expressions in all samples [47]. Quantification of selected gene expression was performed using the comparative threshold cycle (2^−ΔΔCT^) method [48]. The experiment was repeated three times.

## 5. Conclusions

In conclusion, our present study generated transcriptome-wide maps of the m6A profiles and distribution patterns of goat intramuscular adipocytes before and after differentiation based on the MeRIP-seq technique. We found that the different m6A-circRNAs might be involved in the differentiation of intramuscular preadipocytes. Meanwhile, the m6A-circRNAs work as molecular sponges for miRNAs and may play essential roles in regulating miRNA target gene expression during goat adipocyte differentiation. Additionally, this study also explores the correlation between m6A methylation and the level of circRNA expression, indicating the m6A-circRNAs may act through a potential regulatory mechanism in promoting the differentiation of goat adipocytes.

## Figures and Tables

**Figure 1 ijms-24-04817-f001:**
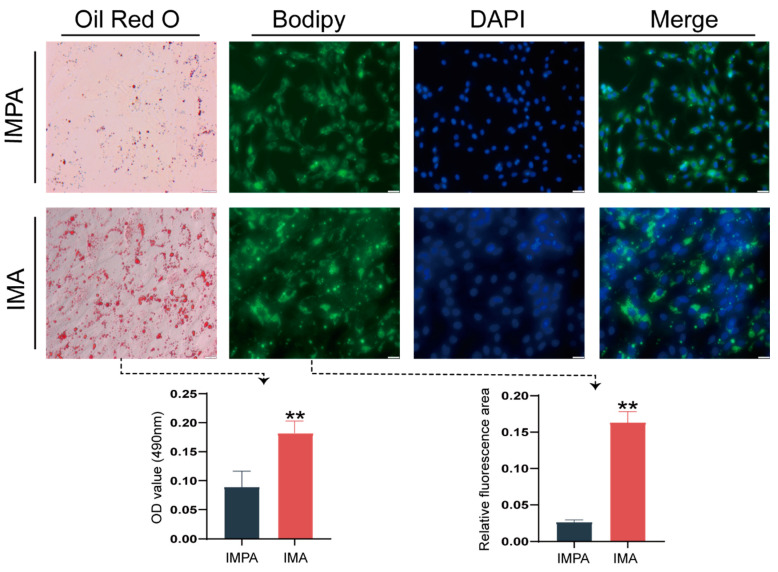
Adipocytes differentiated for 3 days. Oil Red O (Scale bars, 50 µm) and BODIPY staining analysis in goat intramuscular adipocyte adipogenesis. Confocal imaging of lipid droplets after staining with 0.2 mM BODIPY 493/503 (green) and DAPI (blue). Data are shown as mean ± SEM, ** *p* < 0.01. Values for each time point are based on three biological replicates and three technical replicates. IMPA represents intramuscular preadipocytes; IMA represents intramuscular adipocytes.

**Figure 2 ijms-24-04817-f002:**
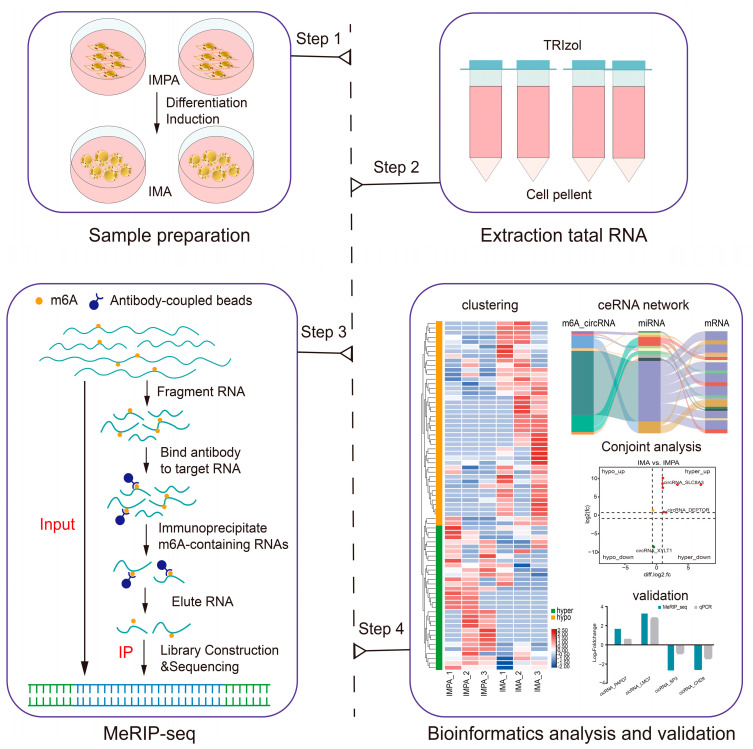
Schematic diagram of the experimental flow. IMAP indicates intramuscular preadipocytes and IMA indicates intramuscular adipocytes.

**Figure 3 ijms-24-04817-f003:**
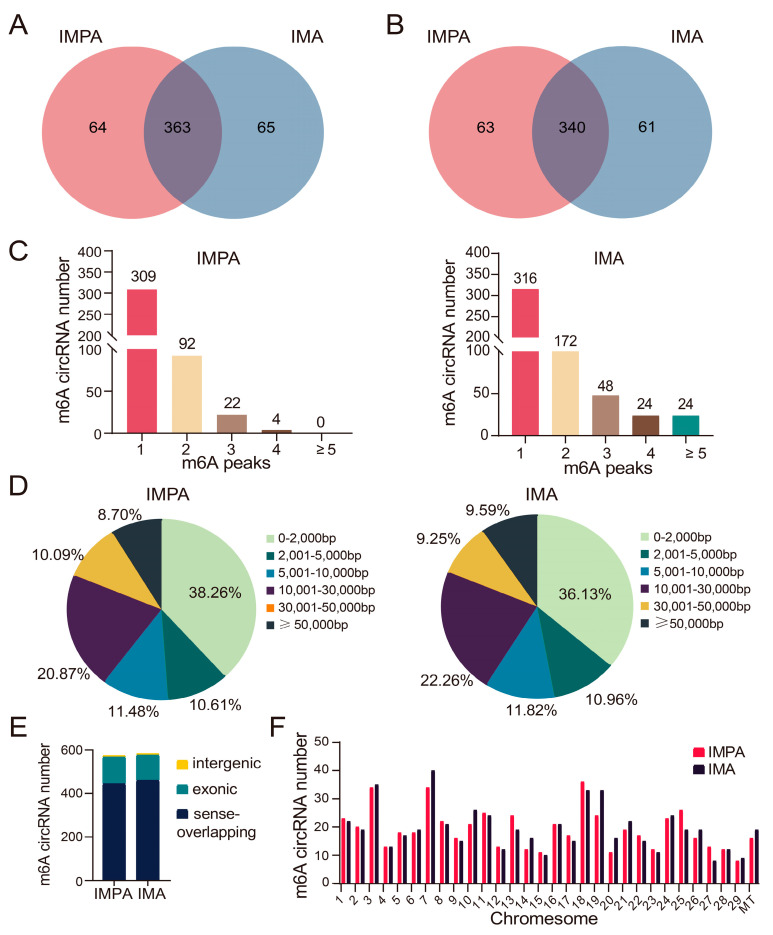
Overview of m6A-circRNAs in the IMPA and IMA groups. (**A**) Venn diagram displaying the specific m6A peaks in the IMPA and IMA groups. (**B**) Venn diagram showing the specific circRNAs with m6A modification between the two groups. (**C**) The number of m6A peaks per circRNA in the two groups. (**D**) The length of m6A-circRNAs in the two groups. (**E**) The source of m6A-circRNAs in the two groups. (**F**) Chromosome distribution.

**Figure 4 ijms-24-04817-f004:**
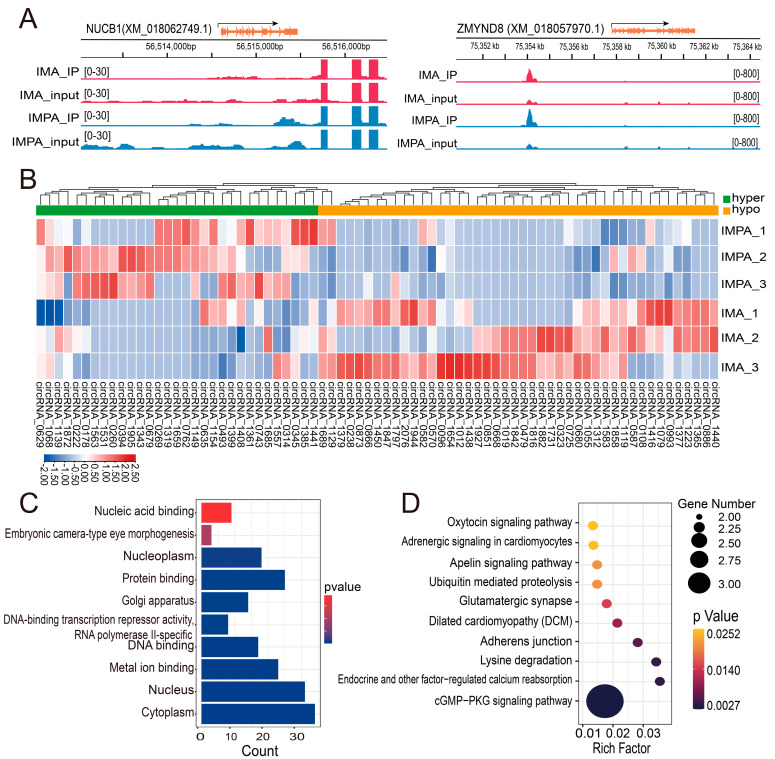
Differential expression of m6A-circRNAs between IMA and IMPA group. (**A**) Data visualization analysis was performed using IGV, showing the location of differential m6A peaks in the circRNA source genes (circRNA_NUCB1 and ZMYND8) between the IMA group and IMPA group. (**B**) Heatmap of m6A-circRNA expression per sample. (**C**) The top 10 GO terms enriched for the source genes of differential m6A-circRNAs in the two groups. (**D**) The top 10 KEGG pathways enriched for the source genes of differential m6A-circRNAs in the two groups.

**Figure 5 ijms-24-04817-f005:**
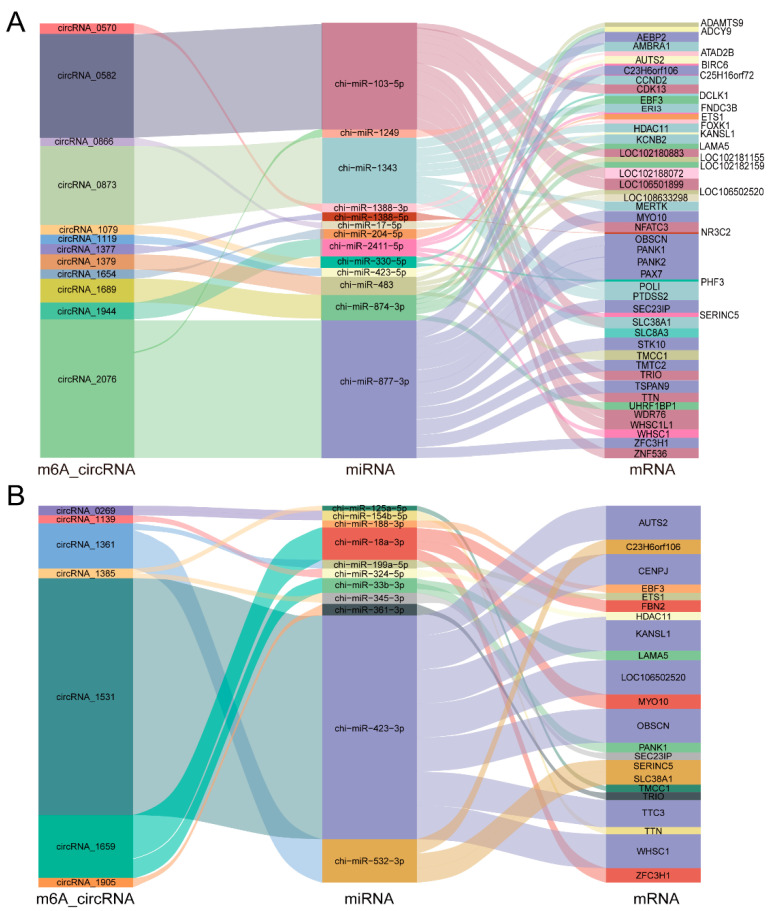
m6A-circRNA–miRNA–mRNA ceRNA interaction network. (**A**) The hypermethylated circRNAs. (**B**) The hypomethylated circRNAs. The box size represents the strength of the binding. The color indicates that the miRNA has a binding site with mRNA and m6A-circRNA.

**Figure 6 ijms-24-04817-f006:**
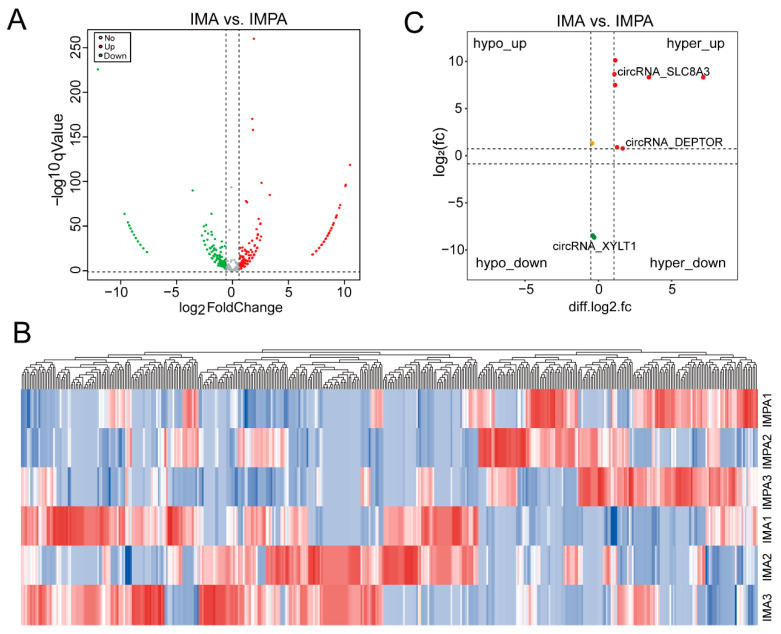
Conjoint analysis of circRNAs with or without m6A modification. (**A**) Volcano plot showing the differentially expressed circRNAs in the IMPA and IMA group. (**B**) Heatmap illustrating the differentially expressed circRNAs in the two groups. (**C**) Conjoint analysis of expression levels and m6A modifications in the IMPA and IMA groups.

**Figure 7 ijms-24-04817-f007:**
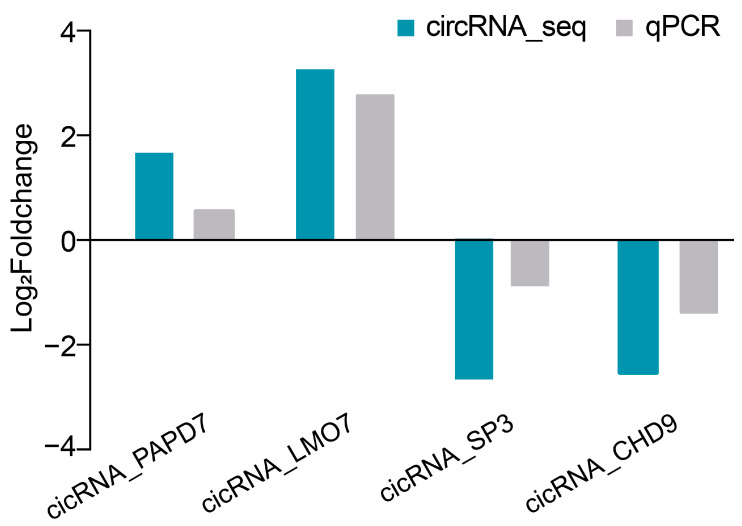
Comparison of circRNA expressions between real-time PCR and circRNA-seq data. For each circRNA, three technical replicates and three biological replicates were used at each sampling time point.

**Table 1 ijms-24-04817-t001:** Output statistics of the sequencing reads for each sample.

Sample	Raw_Reads(M)	Clean_Reads(M)	Clean_Bases(G)	Valid_Bases(%)	Q30(%)	GC(%)
IMA1	50.38	49.57	14.17	93.75	94.08	60.94
IMA2	50.11	49.27	14.05	93.47	93.83	60.63
IMA3	50.3	49.49	14.22	94.24	93.82	60.74
IMPA1	49.83	49.01	13.99	93.58	94.03	61.13
IMPA2	49.37	48.62	14.02	94.66	94.02	60.68
IMPA3	50.31	49.49	14.11	93.49	93.89	61.01
IMA1_input	47.97	47.05	13.23	91.94	95.93	57.21
IMA2_input	49.37	48.39	13.51	91.21	96.04	56.96
IMA3_input	47.29	46.34	12.98	91.49	96.1	57.41
IMPA1_input	47.73	46.82	13.16	91.9	95.94	57.5
IMPA2_input	48.35	47.44	13.31	91.77	95.99	56.77
IMPA3_input	47.88	46.89	13.07	90.99	95.87	57.06

**Table 2 ijms-24-04817-t002:** Summary of the clean read alignment to the goat reference genome.

Sample	Total_Reads	Total_Mapped	Multiple_Mapped	Uniquely_Mapped
IMA1	99,139,058	95,401,484 (96.22%)	4,848,347 (4.89%)	90,553,137 (91.33%)
IMA2	98,545,924	94,865,327 (96.26%)	4,698,299 (4.76%)	90,167,028 (91.49%)
IMA3	98,984,968	95,084,400 (96.05%)	5,016,142 (5.06%)	90,068,258 (90.99%)
IMPA1	98,018,414	94,268,208 (96.17%)	5,241,506 (5.34%)	89,026,702 (90.82%)
IMPA2	97,232,168	93,538,409 (96.20%)	4,702,985 (4.83%)	88,835,424 (91.36%)
IMPA3	98,974,940	95,109,577 (96.09%)	5,017,553 (5.06%)	90,092,024 (91.02%)
IMA1_input	94,096,060	89,868,600 (95.50%)	10,729,437 (11.40%)	79,139,163 (84.10%)
IMA2_input	96,775,848	92,699,532 (95.78%)	10,800,206 (11.16%)	81,899,326 (84.62%)
IMA3_input	92,682,072	88,584,369 (95.57%)	10,356,897 (11.17%)	78,227,472 (84.40%)
IMPA1_input	93,638,328	89376,862 (95.44%)	10,923,483 (11.66%)	78,453,379 (83.78%)
IMPA2_input	94,873,142	90,865,112 (95.77%)	10,783,550 (11.36%)	80,081,562 (84.40%)
IMPA3_input	93,789,268	89,386,330 (95.30%)	10,808,685 (11.52%)	78,577,645 (83.78%)

**Table 3 ijms-24-04817-t003:** The top 10 differential m6A methylation peaks between the IMA and IMPA groups.

**Chr**	**Start**	**End**	**circRNA**	**Log_2_FC**	** *p* ** **-Value**	**Regulation**	**Gene Name**
Chr12	35,549,676	35,551,085	circRNA_1055	3.32478924	3.12E-87	Up	*LMO7*
Chr18	56,511,885	56,517,614	circRNA_1438	9.255672903	1.11E-62	Up	*NUCB1*
Chr13	33,128,553	33,129,648	circRNA_1119	2.485573285	5.46E-55	Up	*ZEB1*
Chr10	21,218,634	21,218,882	circRNA_0873	8.626127066	2.59E-44	Up	*SLC8A3*
Chr10	16,161,014	16,161,210	circRNA_0866	2.253712935	1.31E-42	Up	*LOC102187597*
Chr3	10,437,366	10,437,666	circRNA_0238	8.487830648	4.97E-41	Up	*ZMYM4*
Chr11	74,783,037	74,799,539	circRNA_1012	8.339699789	9.02E-38	Up	*ATAD2B*
Chr27	12,234,705	12,235,515	circRNA_1944	8.329999645	1.44E-37	Up	*WHSC1L1*
Chr13	61,360,331	61,366,664	circRNA_1079	8.30930326	3.91E-37	Up	*DCLK1*
Chr20	66,412,450	66,418,587	circRNA_1583	1.733511924	3.03E-35	Up	*PAPD7*
Chr18	23,020,758	23,021,057	circRNA_1399	2.578567264	1.12E-51	Down	*CHD9*
Chr13	75,314,614	75,354,619	circRNA_1149	9.125673331	3.81E-49	Down	*ZMYND8*
Chr19	53,784,636	53,812,138	circRNA_1531	8.986718904	1.48E-45	Down	*SEPTIN9*
Chr18	3,010,918	3,011,590	circRNA_1385	2.205131135	2.40E-43	Down	*DDX19A*
Chr2	113,508,380	113,508,876	circRNA_0178	2.756039782	4.60E-41	Down	*SP3*
Chr26	11,981,485	11,988,759	circRNA_1905	8.661060515	4.30E-38	Down	*SEC23IP*
Chr8	40,220,320	40,221,369	circRNA_0743	2.262451821	3.39E-36	Down	*GLIS3*
Chr8	76,948,426	76,949,174	circRNA_0762	8.473007382	2.47E-34	Down	*KIF27*
Chr20	13,792,302	13,792,500	circRNA_1563	8.471672577	2.62E-34	Down	*NLN*
Chr17	9,726,962	9,729,487	circRNA_1343	8.469304477	2.91E-34	Down	*LOC102190983*

**Table 4 ijms-24-04817-t004:** The primer information for qPCR.

Gene Name	Primer Sequence	TM/°C	Product Length
*UXT*	GCAAGTGGATTTGGGCTGTAAC	60	180
	ATGGAGTCCTTGGTGAGGTTGT	60	
circRNA_LMO7	TCAAAGTTAGTGTCTGGCAATG	55.7	227
	GTGCTGGACTTTTGTGGG	55.6	
circRNA_PAPD7	TACATCCCAGCACCTAACC	52.9	180
	CATCGGTCTGTTCCATCC	53	
circRNA_CHD9	ACTGCCAGTTCCCGTGACA	59	235
	GAAGGGGTTCGTAGCAGCG	60.6	
circRNA_SP3	GGGTCCTTGTGGGGCTTAC	59	237

## Data Availability

The raw sequence data reported in this paper have been deposited in the Genome Sequence Archive (Genomics, Proteomics & Bioinformatics 2021) in the National Genomics Data Center (Nucleic Acids Res 2022), China National Center for Bioinformation/Beijing Institute of Genomics, Chinese Academy of Sciences (GSA: CRA008981) and are publicly accessible at https://ngdc.cncb.ac.cn/gsa (accessed on 20 November 2022).

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
