# Peer review of "The Important Role of m6A-Modified circRNAs in the Differentiation of Intramuscular Adipocytes in Goats Based on MeRIP Sequencing Analysis"

_ijms, 2023, doi:10.3390/ijms24054817_

Round 1

Reviewer 1 Report

Review Comments

In the current study titled “The important role of m6A modified circRNAs in the differentiation of Intramuscular adipocyte in goats based on MeRIP sequencing analysis”, the authors screened for m6A-modified circRNA in goat preadipocytes and adipocytes using MeRIP Seq technology. The authors identified 75 m6A methylation peaks (in 75 circRNAs) as differentially expressed with 44 hypermethylated peaks (in 44 circRNAs) and 31 hypomethylated peaks (in 31 circRNAs) between IMPA and IMA groups. The KEGG enrichment identified PKG signaling pathway, endocrine and other factor-regulated calcium reabsorption, and lysine degradation. Furthermore, m6A-circRNA–miRNA-mRNA interaction networks predicted the potential m6A-circRNA regulation mechanism in goat adipocytes. The manuscript is interesting, but I have some concerns:

1.     The authors used three samples for their study. How do the authors justify the power of their study?

2.     Any specific reason for using passage 3 for the experiments?

3.     Prior to the experiments, a viability test using an MTT assay is recommended to assess the maximum concentration of fatty acids (oleic acid) on the viability of the cells. How did the authors conclude the concentration of oleic acid (50 μm/L) to be used for the experiment?

4.     The cells were harvested on 0 and 3 days after differentiation induction. Did the authors collect other time points, such as day 6 and day 8? Why did the authors choose to include two points for this study?

5.     The study design is a bit confusing. To my understanding, the authors carried out the differentiation process using an oleic acid induction solution in 3 samples. After induction, the cells were harvested on day 0 and day 3. So, for the study, the authors have D0 (within 24h of incubation with oleic acid) – samples 1, 2 and 3, and D3 (mature adipocytes) – samples 1, 2 and 3. However, the control samples with just the BSA and no fatty acids are missing from each set of experiments. Please justify.

6.     The Oil Red O staining and BODIPY staining was used to distinguish mature adipocytes from preadipocytes during the process of culture. How many samples were used for this analysis? Please mention this in the methodology or results section.

7.     In line 122, the authors mentioned the threshold for differential expression circRNA screening as log2FC ≥ 0.58, while in line 124, the thresholds were log2FC ≥ 1.5. Please check.

8.     The m6A-circRNAs/miRNA interactions were predicted using RNAhybrid. However, I do not see minimum free energy prediction between the RNA sequence and miRNA in the entire manuscript. Please check and include the results. Also, elaborate on how the authors predicted the miRNAs.

9.     How many samples were used for RT-PCR validation? How was the data analyzed and statistical analysis performed?

10.  Before analyzing the sequencing data, how were the adapters removed?

11.  The figure numbers are swapped for Figures 1 and 2 in the additional image folder uploaded with the manuscript.

12.  The authors identified differentially expressed m6A-modified circRNAs in section 3.3. Please include the complete list of differential m6A methylation peaks between the IMA and IMPA groups in the supplementary section.

13.  Cite the tools used to plot Figure 5 in the appropriate section.

14.  In the discussion section, the authors commented on the pathways regulated by adipocyte differentiation; however, the circRNAs validated by RT-PCR are not discussed in the manuscript. Although it is not possible to discuss all the m6A-circRNAs and respective miRNAs, it would be informative to include the biological mechanism of the ones that are validated and were identified as significant.  

15.  Add abbreviation M6A with N6-Methyl-adenosine in line 14

16.  Please check for grammatical mistakes and typos in the entire manuscript. Some of them are listed below:

Line 194 – We finding

Line 208 – Goat Different Adipocytes

Line 216 – shown blow

Line 230 and 231 – Check font size

Line 260 – TOP 10

Italicize the gene names

Author Response

Dear Editor,

We are very grateful to the reviewers who commented on our manuscript with their patience, earnestness, and unique insights to help improve the quality of the manuscript (ID: ijms-2146702). We have revised all issues one by one, as the editor requested. We also responded below to each comment. Hopefully, our manuscript can be further preceded for the peer-review process. The following are the replies to related questions:

With best regards

Yours sincerely

Corresponding author Dr. Yaqiu Lin

In the current study titled “The important role of m6A modified circRNAs in the differentiation of Intramuscular adipocyte in goats based on MeRIP sequencing analysis”, the authors screened for m6A-modified circRNA in goat preadipocytes and adipocytes using MeRIP Seq technology. The authors identified 75 m6A methylation peaks (in 75 circRNAs) as differentially expressed with 44 hypermethylated peaks (in 44 circRNAs) and 31 hypomethylated peaks (in 31 circRNAs) between IMPA and IMA groups. The KEGG enrichment identified PKG signaling pathway, endocrine and other factor-regulated calcium reabsorption, and lysine degradation. Furthermore, m6A-circRNA–miRNA-mRNA interaction networks predicted the potential m6A-circRNA regulation mechanism in goat adipocytes. The manuscript is interesting, but I have some concerns:

Point 1: The authors used three samples for their study. How do the authors justify the power of their study?

Response 1: Thanks for your reminder. We have made relevant modifications.

Point 2: Any specific reason for using passage 3 for the experiments?

Response 2 : Thank you for your suggestion. We used the 3rd generation cells because we found that the induction differentiation efficiency of goat intramuscular adipocytes was high in the 0-3rd generation, but decreased after the 4th and 5th generation, so we chose the 3rd generation.

Point 3: Prior to the experiments, a viability test using an MTT assay is recommended to assess the maximum concentration of fatty acids (oleic acid) on the viability of the cells. How did the authors conclude the concentration of oleic acid (50 μm/L) to be used for the experiment?

Response 3: Thank you for your valuable comments. When the intramuscular precursor adipocyte culture system of meat goats was established in the laboratory, the concentration of oleic acid induction solution was screened and 50 μm/L was finally determined as the optimal concentration, which was adopted in the related articles published in the laboratory recently (Xiong et al., 2018; Du et al.,2021; Li et al.,2022).

[1] Xiong Y, Xu Q, Lin S, Wang Y,Lin YQ, Zhu JJ. Knockdown of LXRα inhibits goat intramuscular preadipocyte differentiation. Int J Mol Sci, 2018, 19(10): 3037.

[2] Du Y, Wang Y, Li YY, Emu QZ, Zhu JJ, Lin YQ. miR-214-5p Regulating Differentiation

of Intramuscular Preadipocytes in Goats via Targeting KLF12. Front Genet, DOI: 10.3389/ fgene.2021.748629.

[3] Li X, Zhang H, Wang Y, Li YY, Wang YL, Zhu JJ, Lin YQ. Chi-Circ_0006511 Positively Regulates the Differentiation of Goat Intramuscular Adipocytes via Novel-miR-87/CD36 Axis. Int J Mol Sci,2022, doi:10.3390/ijms232012295.

Point 4: The cells were harvested on 0 and 3 days after differentiation induction. Did the authors collect other time points, such as day 6 and day 8? Why did the authors choose to include two points for this study?

Response 4: Thank the experts for their valuable comments. In our laboratory's previous study, it was found that oleic acid had reached nearly 100% differentiation at 5 days of induction, so we did not choose 6 days and 8 days. The aim of this experiment is to screen the differential circRNA and m6A-circRNA of intramuscular adipocytes of meat goats before and after differentiation. In this experiment, intramuscular adipocytes of 0 and 3 days of induced differentiation were selected. Figure 1 in the article proved that the intramuscular adipocytes of 3 days of induction differentiation were in a differentiated state through morphological staining and OD value measurement.

Point 5: The study design is a bit confusing. To my understanding, the authors carried out the differentiation process using an oleic acid induction solution in 3 samples. After induction, the cells were harvested on day 0 and day 3. So, for the study, the authors have D0 (within 24h of incubation with oleic acid) – samples 1, 2 and 3, and D3 (mature adipocytes) – samples 1, 2 and 3. However, the control samples with just the BSA and no fatty acids are missing from each set of experiments. Please justify.

Response 5: Thank you for your valuable comments. In this study, in order to screen the differential circRNA and m6A-circRNA before and after the differentiation of intramuscular adipocytes in meat goats, we selected two groups, one group is the precursor adipocytes without induction (D0 in this paper), and the other group is the adipocytes that have been induced by oleic acid for 3 days. Thank you again for your suggestion. If necessary, we will consider your proposal in the follow-up experiment to further improve the research content.

Point 6: The Oil Red O staining and BODIPY staining was used to distinguish mature adipocytes from preadipocytes during the process of culture. How many samples were used for this analysis? Please mention this in the methodology or results section.

Response 6: Thanks for your valuable comments. Oil red O staining and BODIPY staining were set based on the number of biological replicates that we sequenced. Three biological replicates and three technical replicates at each time point. We have modified it according to your suggestions (Line 147).

Point 7: In line 122, the authors mentioned the threshold for differential expression circRNA screening as log2FC ≥ 0.58, while in line 124, the thresholds were log2FC ≥ 1.5. Please check.

Response 7: Thank you for your comments. We have modified it according to your suggestions.

Point 8: The m6A-circRNAs/miRNA interactions were predicted using miRanda. However, I do not see minimum free energy prediction between the RNA sequence and miRNA in the entire manuscript. Please check and include the results. Also, elaborate on how the authors predicted the miRNAs.

Response 8: Thank you for your suggestion. we have added the information required as explained above the minimum free energy prediction between the RNA sequence and miRNA in the Results section (Line 226-227), Associations identified with m6A-circRNAs/miRNA interactions are shown in Table S2. For miRNAs predictions, please see it in the Materials and Methods section (Line 120-121).

Point 9: How many samples were used for RT-PCR validation? How was the data analyzed and statistical analysis performed?

Response 9: Thanks for your suggestion. When RT-PCR validation, for each circRNA, three technical replicates and three biological replicates were used at each sampling time point. We have added the information required as explained above (Line 277-279). For statistical analysis, please see it in the Materials and Methods section (line 124-126).

Point 10: Before analyzing the sequencing data, how were the adapters removed?

Response 10: Thank you for your suggestion. In this paper, through the Illumina platform, a large amount of sample paired end sequencing data was obtained. Given the impact of data error rate on the results, fastp software for quick quality control to see if adapter removal was necessary and to ensure that sequenced data was of sufficient quality.

Point 11: The figure numbers are swapped for Figures 1 and 2 in the additional image folder uploaded with the manuscript.

Response 11: Thank you for your reminder. This has been revised as suggested.

Point 12: The authors identified differentially expressed m6A-modified circRNAs in section 3.3. Please include the complete list of differential m6A methylation peaks between the IMA and IMPA groups in the supplementary section.

Response 12: Thank you for your valuable comments. We have modified it according to your suggestions. As detailed in Table S1.

Point 13: Cite the tools used to plot Figure 5 in the appropriate section.

Response 13: Considering the Reviewer’s suggestion, we have added the information required as explained above (Line 122-123).

Point 14: In the discussion section, the authors commented on the pathways regulated by adipocyte differentiation; however, the circRNAs validated by RT-PCR are not discussed in the manuscript. Although it is not possible to discuss all the m6A-circRNAs and respective miRNAs, it would be informative to include the biological mechanism of the ones that are validated and were identified as significant.

Response 14: We are very grateful to the Reviewer for reviewing the paper so carefully. We completed the addition of circRNAs for RT-PCR validation in the discussion section and adjusted the text where highlighted (line 360-366).

Point 15: Add abbreviation M6A with N6-Methyl-adenosine in line 14

Response 15: This has been revised as suggested.

Point 16: Please check for grammatical mistakes and typos in the entire manuscript. Some of them are listed below:

Line 194 – We finding

Line 208 – Goat Different Adipocytes

Line 216 – shown blow

Line 230 and 231 – Check font size

Line 260 – TOP 10

Italicize the gene names

Response 16: Thank you for your reminder. This has been revised as suggested.

Reviewer 2 Report

The research work is interesting and the manuscript is well written.

Since the effect of circRNAs on mRNA expression is very well known. I wonder why wasn't RNAseq performed on this dataset. It would be valuable to include that without which nothing much about the possible regulation of circRNA can be inferred with certainty.

Author Response

Dear Editor,

We are very grateful to the reviewers who commented on our manuscript with their patience, earnestness, and unique insights to help improve the quality of the manuscript (ID: ijms-2146702). We have revised all issues one by one, as the editor requested. We also responded below to each comment. Hopefully, our manuscript can be further preceded for the peer-review process. The following are the replies to related questions:

With best regards

Yours sincerely

Corresponding author Dr. Yaqiu Lin

Point 1: The research work is interesting and the manuscript is well written.

Since the effect of circRNAs on mRNA expression is very well known. I wonder why wasn't RNAseq performed on this dataset. It would be valuable to include that without which nothing much about the possible regulation of circRNA can be inferred with certainty.

Response 1: First, thank the reviewers for their affirmation of our work. Whole transcriptome sequencing was used in our study, which provides a wide variety of applications from mRNA profiling to analyze of the entire transcriptome analysis including mRNA and non-coding RNA (lncrna, circrna). However, in this study, we focused on the effect of m6A-modified circRNA on intramuscular adipocytes of goats before and after differentiation.

Round 2

Reviewer 1 Report

Thank you for the edits. Please modify the manuscript with few more edits in this version.

1.     In the previous version of manuscript, the authors mentioned the sample size in section 2.1. It is removed from the current version of manuscript. Line 105 mentions that total RNA from 12 samples were extracted. This gives the impression that the authors used 12 different samples until the readers identify it from figure legend. The authors are requested to mention the samples used in the study as n = 3 in section 2.1. But please justify the power of study for the respective experiments.

2.     Please check the reference style mentioned by the journal. The manuscript does not follow the citation style as required by the journal,

3.     Italicize the gene name UXT in line 131

4.     There are two Table 1 in the manuscript. Please modify the table numbers.

5.     Line 116 – “The DAVID database was used to conduct GO enrichment analysis [50]. KOBAS [51]”. What is [50] and [51] in this phrase? There are no references as 50 and 51.

6.     Line 171 - We used circRNA-seq to compare the circRNA of differences between intramuscular. Correct the sentence as “We used circRNA-seq to compare the differences in circRNAs between intramuscular preadipocytes (IMPA) and adipocytes (IMA).

7.     Line 230 – Complete the sentence “As detailed in Table S2”

Author Response

Dear Editor,

We are very grateful to the Editor and the reviewers who commented on our manuscript with their patience, earnestness, and unique insights to help improve the quality of the manuscript (ijms-2146702). We have revised all issues one by one, as the editor requested. We also responded below to each comment. Hopefully, our manuscript can be further preceded for the peer-review process. The following are the replies to related questions:

With best regards

Yours sincerely

Corresponding author Dr. Yaqiu Lin

Response to Reviewer 1 Comments

Thank you for the edits. Please modify the manuscript with few more edits in this version.

Point 1: In the previous version of manuscript, the authors mentioned the sample size in section 2.1. It is removed from the current version of manuscript. Line 105 mentions that total RNA from 12 samples were extracted. This gives the impression that the authors used 12 different samples until the readers identify it from figure legend. The authors are requested to mention the samples used in the study as n = 3 in section 2.1. But please justify the power of study for the respective experiments.

Response 1: Thank you very much for reviewing our manuscript. In this study, three longissimus dorsi muscles of 7-day-old Jianzhou Daer goats were collected and cultured intramuscular adipocytes cells. Briefly, the cells of each sheep were duplicated in multiple petri dishes, and eventually assembled into a sample, which was subsequently sequenced. This met both biological and technical replicates, while also achieving the company's loading capacity. In addition, total RNA from 6 samples were extracted, which has been modified in the text (Line 95).

Point 2: Please check the reference style mentioned by the journal. The manuscript does not follow the citation style as required by the journal,

Response 2: Thank you for your comments. We have modified it according to your suggestions.

Point 3: Italicize the gene name UXT in line 131

Response 3: Thanks for your reminder. We have modified it according to your suggestions.

Point 4: There are two Table 1 in the manuscript. Please modify the table numbers.

Response 4: Thank you for your suggestion. We have revised this mistake.

Point 5: Line 116 – “The DAVID database was used to conduct GO enrichment analysis [50]. KOBAS [51]”. What is [50] and [51] in this phrase? There are no references as 50 and 51.

Response 5: Thank you for your valuable comments. We have revised this mistake and re-added the reference.

Point 6: Line 171 - We used circRNA-seq to compare the circRNA of differences between intramuscular. Correct the sentence as “We used circRNA-seq to compare the differences in circRNAs between intramuscular preadipocytes (IMPA) and adipocytes (IMA).

Response 6: We appreciate it very much on this constructive suggestion, and we have revised it according to your ideas.

Point 7: Line 230 – Complete the sentence “As detailed in Table S2”

Response 7: Thanks for your reminder. We have re-descripted the sentence to make it clearer. (Line 230-231)